

# Improving patient rehabilitation performance in exercise games using collaborative filtering approach

Waidah Ismail[1,2], Ismail Ahmed Al-Qasem Al-Hadi[1,3], Crina Grosan[4] and Rimuljo Hendradi[2]

[1] Faculty of Science and Technology, Universiti Sains Islam Malaysia, Nilai, Negeri Sembilan, Malaysia
[2] Information System Study Program, Faculty of Science and Technology, Universitas Airlangga, Indonesia Kampus C, Surabaya, Indonesia
[3] Faculty of Ocean Engineering Technology and Informatics, Universiti Malaysia Terengganu, Kuala Nerus, Terengganu, Malaysia
[4] Department of Computer Science, Brunel University, London, United Kingdom

## ABSTRACT

**Background:** Virtual reality is utilised in exergames to help patients with disabilities improve on the movement of their limbs. Exergame settings, such as the game difficulty, play important roles in the rehabilitation outcome. Similarly, suboptimal exergames' settings may adversely affect the accuracy of the results obtained. As such, the improvement in patients' movement performances falls below the desired expectations. In this paper, a recommender system is incorporated to suggest the most preferred movement setting for each patient, based on the movement history of the patient.

**Method:** The proposed recommender system (ResComS) suggests the most suitable setting necessary to optimally improve patients' rehabilitation performances. In the course of developing the recommender system, three methods are proposed and compared: ReComS (K-nearest neighbours and collaborative filtering algorithms), ReComS+ ($k$-means, K-nearest neighbours, and collaborative filtering algorithms) and ReComS++ (bacterial foraging optimisation, $k$-means, K-nearest neighbours, and collaborative filtering algorithms). The experimental datasets are collected using the Medical Interactive Recovery Assistant (MIRA) software platform.

**Result:** Experimental results, validated by the patients' exergame performances, reveal that the ReComS++ approach predicts the best exergame settings for patients with 85.76% accuracy.

# INTRODUCTION

Human disabilities could develop through cerebral palsy (CP), stroke, spinal cord injury (SCI), traumatic brain injury (TBI), and humerus fracture (*Tousignant et al., 2014*). Rehabilitating these disabled patients can be achieved using traditional and modern treatments. Virtual reality (*Turolla et al., 2013*), robotics (*Díaz, Gil & Sánchez, 2011*; *Maciejasz et al., 2014*), simulation, and exergames (*Covarrubias et al., 2015*) are commonly

Corresponding author
Waidah Ismail,
waidah@usim.edu.my

used in modern treatments. These games are quite promising as they furnish patients with new experiences while performing their daily exercises which are rehabilitation therapies (*Li et al., 2018*). Rehabilitation is traditionally based on the assessment requirements drawn through physiotherapy. Rehabilitation therapy and assessment are provided by rehabilitation centres, where patients train their disabled limbs through a series of pre-determined exercises. Such procedures help to improve the movement of the limbs and improve their functionality.

One of the recently developed systems for training lower limbs is the robotic system. Currently, there are five types: foot-plate-based gait trainers, treadmill gait trainers, overground gait trainers, active foot orthoses, in addition to stationary gait and ankle trainers (*Díaz, Gil & Sánchez, 2011*). These systems include passive robotic devices, which assist the patient to train the idle limb(s) of the lower part. The virtual reality therapy (VRT) systems involve the use of virtual reality as assistant-tools for the rehabilitation process. In assistive technology, serious games come with multimodal functions and immersive characteristics that have been embedded into various types such as robotics, virtual reality, and simulator. Thus, serious games are promising technology that can bring new experiences for people with disabilities to perform their rehabilitation (*Li et al., 2018*; *Merilampi et al., 2017*). In addition, exergames are VRT systems that are used by patients who suffer from movement disability in their idle limbs. Through several training procedures, exergames help patients to improve on the physical movements of their muscles (*Jaarsma et al., 2020*) so that they can move their idle parts gradually. For example, the VRT system employs three exergames (the bike, the pedal boat, and the swimmer) that consists of a virtual system comprising a strengthening machine, Kinect device, large screen, and a computer (*Pruna et al., 2018*). As for the spine, the VRT system is a non-invasive alternative having minimal negative or harmful effects (*Chi et al., 2019*). The VRT systems for lower limbs and spine; however, remain inefficient as existing VRT systems only cater to the upper limbs.

Exergame therapy forms part of the rehabilitation approaches that are offered to patients in rehabilitation centres. Exergames refer to video games (*Da Gama et al., 2016*) that encourage patients to continue their exercise without feeling bored. The therapy consists of an iteration of exercises that focus mainly on strengthening a part of the patient's body such as the knee. For effective results, it is essential that the patient performs the right movement following the rules of each exergame; otherwise, the benefits may not be pronounced and the desired results will be less noticeable (*Da Gama et al., 2016*). Different devices provide the virtual environment based on the requirements of the VRT application. These include a large monitor, virtual interface (*Brokaw & Brewer, 2013*), Microsoft Kinect, Xbox (*Baur et al., 2018*), strengthening machine (*Pruna et al., 2018*), customised metal rig that holds standard wheelchair and robotic devices (*Radman, Ismail & Bahari, 2018*).

The Medical Interactive Recovery Assistant (MIRA) platform is a new VRT application that presents a wide variety of games and movements for various rehabilitation needs. It consists of three parts: adapted movement-based interactive video games, the Kinect, and the leap motion sensors (*Moldovan et al., 2017*). The Kinect sensor tracks motion and

provides different interactions between the patient and the different types of exergames (*Mcglinchey et al., 2015*). Leap motion tracks the hand's movement that composed with the flexion gauges placed into the glove (*Borja et al., 2018*). In other words, the exergames in MIRA are created specifically to aid physical rehabilitation therapies and assessments. An example is that studies the movement performance of children who are 7 years old and suffer from the brachial plexus palsy caused by transverse myelitis (*Czakó, Silaghi & Vizitiu, 2017*). In the study, the movement performance of children was improved by MIRA exergames training. Another case study demonstrates that MIRA exergames have positive effects and can be safely implemented for adult patients (*Mcglinchey et al., 2015*). However, in both case studies (i.e., *Mcglinchey et al., 2015*; *Czakó, Silaghi & Vizitiu, 2017*), a physiotherapist uses default exergames settings and no automation was considered. Prediction scoring method is used to suggest the comfortable difficulty mode for rehabilitation patients using k-means algorithm (*Zainal et al., 2019*). It is used to analyse five variables that are generated by patients when playing MIRA exergames. However, there are various exergames with several variables needed to be analysed for finding the accurate prediction scoring.

In the above scenarios, the most suitable exergame settings are required considering each patient's disability type. As such, due to the low engagement between physiotherapists and patients, physiotherapists use default settings which invariably lowers the accuracy in playing the exergames and reduces the patients' performances. To overcome this problem, a recommender system (RS) is needed in the MIRA platform. However, to the best of our knowledge, addressing this problem using RS is absent in the literature. In addition, the decision tree model applied to predict a patient's rehabilitation future performance is based on time, average acceleration, distance, moving time, and average speed. This prediction method uses the default exergame settings and the previous performances of patients who played the same exergame with the same side (*Zainal et al., 2020*). In spite of that, the prediction will be more accurate if the exergame settings are controlled automatically. Therefore, RS is utilised in this paper to suggest appropriate settings for patients who use MIRA to enhance their movement disabilities.

RS is a subcategory of an information filtering system that aims to forecast or project the "rating" or "preference" of a person (*Ismail et al., 2019*). Over the last 10 years, RSs have been explored and used in various applications that include e-health, e-learning, e-commerce, and knowledge management systems (*Xu, Zhang & Yan, 2018*; *Zainal et al., 2020*). In the same manner, we deploy the RS in this research. Traditionally, an exergame records a patient's information and performance during a session. The recorded data is used to monitor the progress of the patient (*Covarrubias et al., 2015*). Likewise, the system analyses patients' movements during the exercise and generates statistical data. Nonetheless, several challenges have been identified while creating tailored exergame schedules for patients. In this respect, this paper explores the use of ReComS as an interface for each exergame, using the patient's movement history as a benchmark. To address the problem related to the input setting, the ReComS approach applies $k$-means, K-nearest neighbours (K-NN), collaborative filtering (CF), and bacterial foraging optimisation algorithm (BFOA) to accurately predict input variables through an item settings dialogue

box in the MIRA platform itself. From the above discussion, the primary contributions of this research include the following:

1. The proposed ReComS approach that suggests the most appropriate setting for enhancing patients' rehabilitation performance.
2. A novel deployment of RS in the MIRA platform to accurately suggest the most suitable settings needed to improve the limb movement of patients.
3. An enhancement of input variables' prediction accuracy using three newly developed methods: ReComS (K-NN and CF algorithms), ReComS+ ($k$-means, K-NN and CF algorithms) and ReComS++ (BFOA, $k$-means, K-NN, and CF algorithms).

The remainder of this paper is structured as follows: "MIRA Platform" describes the MIRA platform. The concepts used in this paper are defined in "Definition of Concepts". In "Materials & Methods", the proposed methods are provided. "Results" and "Conclusion" present the empirical results and conclusion, respectively.

## MIRA PLATFORM

The MIRA platform is an effective system that allows patients to play their way towards recovery (*Zainal et al., 2019*). MIRA is a non-immersive type of VRT application developed to make physiotherapy entertaining and enjoyable for patients. The platform transforms prevailing physical therapy exercises into clinically-designed video games. Asides from improving patients' interests in exercising, an external sensor monitors and evaluates their adherence. MIRA contains a broad range of games and exercises for the upper limbs, lower limbs, and spine. Fig. 1 illustrates three instances of games in the MIRA platform. First, the patient plays the Catch game with hip abduction movement (Fig. 1A). In the second image, the patient plays the Airplane game with elbow flexion in abduction movement (Fig. 1B). In the third example, the patient plays the Flight control game with general shoulder movement (Fig. 1C). All MIRA games are played following the rules of each game and movement.

The Kinect motion sensor camera and screens are connected to computers (as shown in Fig. 1) to provide the virtual environment. The physiotherapist utilises these devices and the MIRA platform to create a session for patients based on their ability and rehabilitation needs. He combines exercises and games with the specific difficulty setting, movement tolerance, and range of movement (*Wilson et al., 2017*) based on his observations from the previous data of each patient. Through these settings, this research examines the input and output attributes of the scheduled games in order to auto-determine input variables for each exergame. Traditionally, the physiotherapist tracks the exergame history of each patient (that reflects the movement threshold of the idle limb) then suggests the values of input settings in the dialogue box for future exergaming. This method of observation is costly and time-consuming because the physiotherapist needs to prepare a manual list of variables to track the patient's history. Thus, most physiotherapists use the default setting for all patients which retards the patients' performances, especially when they play using their idle limbs. In view of the aforementioned, this research proposes ReComS approaches

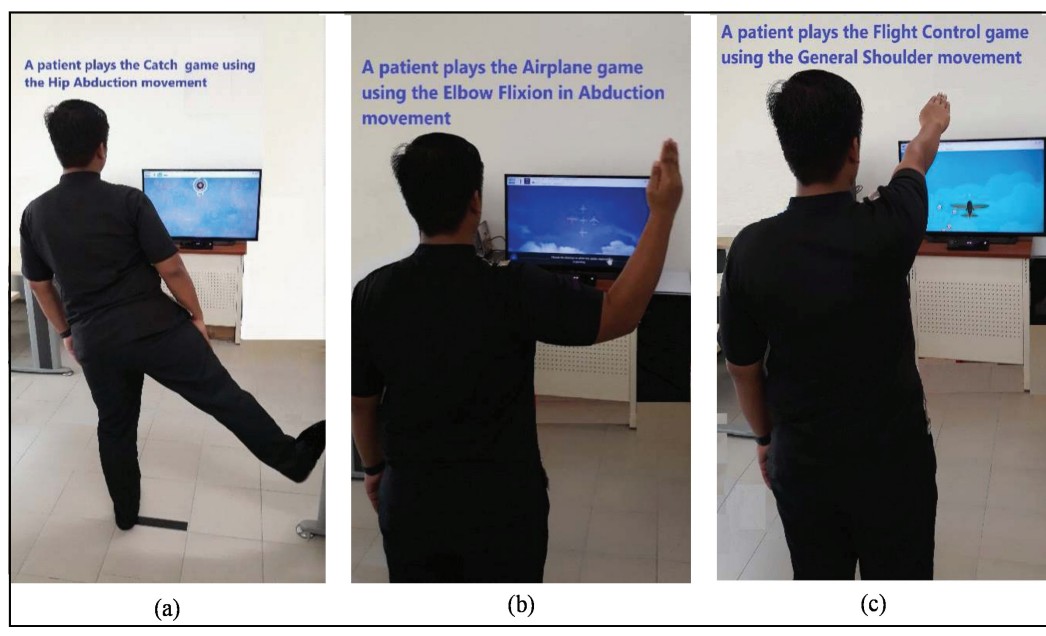

**Figure 1 The case of a patient playing three game-based exercises using the MIRA Platform (A–C).**

for learning the "best" setting for each patient. It deploys $k$-means, K-NN, and BFOA algorithms, in addition to the CF technique. The main reasons for using MIRA data include:

- MIRA data contains several exergame features that can be performed by moving the upper or lower limbs.
- Patients' data are normalized and arranged in a matrix of features. However, this matrix includes a high percentage of zeros (i.e., unknown values) because the exergame output variables are different due to patients' diverse movement skills.
- Some exergames generate a small number of records because only a few patients are interested in playing them.

Our approach is discussed further in the following subsections.

# DEFINITION OF CONCEPTS

## Collaborative filtering (CF)

The use of CF (a popular technique in RSs) to send personalised recommendations to users based on their behaviour has become widely used due to its efficiency in recommending preferred items. CF technique utilises product ratings provided by a collection of customers and recommends products that the target customer has not yet considered but will likely enjoy (*Al-Hadi et al., 2020a*). The rating score (with a value between 1 and 5) is used to indicate if a person likes a product or otherwise. These values are arranged in a matrix as rows. Thereafter, the similarity values between the target customer and other customers in the matrix are calculated to predict the customer's

| Algorithm 1 Pseudocode of the CF technique. |
| --- |

*Input:*

*Matrix features (patients, features)*

*Output:*

*A set of prediction scores for features*

*Steps:*

- *Choose the target patient from the matrix features.*
- *Assign the similarity between target patient and other patients utilising similarity distance.*
- *Assign the prediction value for each feature using the prediction approach.*
- *Measure the accuracy prediction performance using error function, such as Root Mean Squared Error (RMSE) (Al-Hadi et al., 2020b).*

interest in the products (*Natarajan et al., 2020*). In this work, the patients represent users and the generated output features of the MIRA platform represent items. The considered MIRA platform offers a vast range of features for several games and movements managed in a single matrix whereas, the preferences of customers are managed in the rating matrix for estimation via the CF technique. This matrix of features necessitates division using the *k*-means algorithm to minimise the prediction errors. Moreover, Algorithm 1 summarizes the procedure involved in the CF technique.

## *k*-means algorithm

*k*-means is a clustering algorithm that is often used in iterative optimisation, given its efficiency (*Zainal et al., 2019*). Algorithm 2 displays the proximity measures of *k*-means which are city block, hamming, cosine, correlation coefficient, and squared Euclidean Distance (ED). The ED between two points is the length of the straight line that connects them. Within the Euclidean plane, the distance between points $(x_1, y_1)$ and $(x_2, y_2)$ can be calculated using Eq. (1) (*Borja et al., 2018*).

$$ED(x_1, y_1, x_2, y) = \sqrt{(x_1 - y_1)^2 + (x_2 - y_2)^2} \tag{1}$$

## K-nearest neighbours algorithm

K-NN is a simple classification method used to analyse a large matrix of features or to provide recommendations (*Weisstein, 2020*). When new data are required to be categorised, the K-NN algorithm computes the distance in values between the target record and other records. These records are ordered based on distance (*Tarus, Niu & Mustafa, 2018*). At the final stage, the first *k* record will be chosen from the ordered list i.e., K-NN. The pseudocode in Algorithm 3 describes the stages of applying K-NN.

## Bacterial foraging optimisation algorithm

Optimisation algorithms have proven effective in several areas including RS (*Al-Hadi et al., 2017*) and healthcare (*Zainal et al., 2020*). For instance, BFOA has been well-embraced in recent RS approaches for providing high accuracy prediction (*Al-Hadi et al., 2020b, 2017*).

**Algorithm 2** The k-means clustering algorithm.

*Input:*

*D = {d1,d2, …,dn} //set of n data items*

*k //number of desired clusters*

*Output:*

*A set of k clusters*

*Steps:*

1. Arbitrarily choose k data items from D as initial centroids.

2. *Repeat*

    a. *Allocate each item d1 to the cluster with the nearest centroids* (Eq. (1)).

    b. *For each cluster, calculate the new mean.*

    c. *Calculate the junction among clusters to keep the k number of clusters.*

***Until*** *convergence criteria are met.*

---

**Algorithm 3** Pseudocode of K-nearest neighbours algorithm.

*Input:*

Matrix Features (Patients, Features)

*Output:*

*A set of neighbours most like the target patient.*

*Steps:*

1. *Choose the target patient from the matrix features.*

2. *Assign the similarity values amid the target patient and other patients by a similarity distance measure, as shown in* Eq. (1).

3. *Sort the patients starting from the lowest distance value to the topmost distance value (from the highest similarity to the lowest similarity).*

4. *Choose k number of patients from the first in the sorted list.*

---

This motivates us to use BFOA in this experiment for learning patients' latent features and for optimising the output prediction. The BFOA is an evolutionary computational algorithm for global optimisation. It is used to classify and learn better convergence (*Amghar & Fizazi, 2017*). For example, in the human intestine, the BFOA conventions the features of *E. coli* bacteria in the foraging procedure and recycles them for universal optimisation to produce effective clarifications for large ranging issues (*Naveen, Sathish Kumar & Rajalakshmi, 2015*). Similarly, BFOA is utilized to learn patients' latent features which are classified as nearest neighbours within the features of the best cluster while other clusters will be neglected. There are three main phases during the swarming evolution of bacteria which are chemotaxis, reproduction, and elimination-dispersal. These phases are described as follows:

### Chemotaxis

During this stage, each bacterium locates rich nutrients and avoids noxious substances. Patients' accurate features represent rich nutrients that can be tracked by learning the

lowest error value throughout the learning iteration. The chemotaxis stage includes three processes: swimming, tumbling, and swarming. The bacterium swims for a certain period and tumbles while using its flagella to change its swimming direction (*Yang et al., 2016*). The direction of movement after a tumble is given in Eq. (2).

$$(j + 1, k, l) = \beta^i(j, k, l) + C_i + \varnothing / \sqrt{\varnothing_i^t \varnothing_i} \tag{2}$$

where $\beta^i$ represents the members of bacteria $i$ (i.e., latent features of patients), $C_i$ is the stage degree in the direction of the tumble, $j$ denotes the index for the sum of chemotactic, $k$ refers to the index for the number of reproductions, and $l$ reflects the index for the sum of elimination-dispersal. Besides, $\varnothing / \sqrt{\varnothing_i^t \varnothing_i}$ is the random unit length direction shown during the swimming phase. In the swarming mechanism, the latent features of a patient release attractant or repellent signals regarding other patient's latent features, as portrayed in Eq. (3).

$$J_{cc}\left(\beta + \beta^i(j, k, l)\right) = \left[\sum_{i=1}^{S} -d_{attract} \exp\left(w_{attract} \sum_{m=1}^{P} \left(\beta_m - \beta_m^i\right)^2\right)\right] + \left[\sum_{i=1}^{S} -h_{repellant} \exp\left(-w_{repellant} \sum_{m=1}^{P} \left(\beta_m - \beta_m^i\right)^2\right)\right] \tag{3}$$

where $d_{attract}$ is the depth of the attractant that can be recycled to establish the immensity of secretion of attractant by a cell value, $w_{attract}$ is the width of the attractant than can be recycled to denote the means by which the chemical cohesiveness of the signal diffuses, $h_{repellant}$ sets the height of the repellent (a propensity to avoid a nearby cell), and $w_{repellant}$ defines the negligible area where the cell is relative to the diffusion of the chemical signal. *S is the number of groups within the patients' latent features*, $P$ denotes the dimension of the search space, $\beta_m$ is the latent features of group number $m$, and $\beta_m^i$ represents latent feature number $i$ in group $m$.

### Reproduction

This phase deals with the feedback (RMSE) value which acts as the fitness value. These values are obtained after training the target patients' features that have been extracted through the current training stage using the $k$-means, K-NN, BFOA and CF methods. The RMSE values will be saved in an array before sorting (smaller and larger values). The lower half of the latent features having a larger fitness value (dies) while the outstanding latent features or the other half of the population is separated into two equivalent parts having equal values. This phase keeps the bacteria population constant. Eq. (4) shows the healthy values for patients' latent features.

$$J_{health}^i = \sum_{j=1}^{N_c+1} J(i, j, k, l), \tag{4}$$

where $i$ is the sum of patients' latent features, $j$ is the sum of chemotactic steps $N_c$, $k$ is the reproduction step, and $l$ is the elimination-dispersal step.

### Elimination-dispersal

This phase provides the position shifting probability for the limited latent features of patients. The random vectors are produced and arranged in ascending order.

## MATERIALS & METHODS

In this study, ReComS, ReComS+ and ReComS++ approaches are proposed to recommend preferences for the MIRA platform. These preferences are used to determine the input settings of the exergames by learning the precise behaviours of patients. ReComS integrates the K-NN and CF methods to classify the predicted values by reducing the error value. The error value is obtained using the projected and actual values of the previous session of the exergame. The ReComS+ approach improves the prediction performance of ReComS by integrating the $k$-means algorithm with K-NN and CF, which reduces the error value. However, this error value is relatively high. Hence, it lowers the prediction performance of the ReComS and ReComS+. ReComS++ further reduces the error value by optimising the prediction or projection values and integrating $k$-means, K-NN, CF, and BFOA algorithms.

### Dataset

This study was carried out in the rehabilitation centre of Melaka, Malaysia to analyse the generated data using MIRA platform with ethic approval no PRPTAR.600-5(27) by Pusat Rehabilitasi Perkeso Sdn. Bhd., Lot PT 7263, Bandar Hijau, Hang Tuah Jaya, 75450 Bemban, Melaka, Malaysia. The MIRA platform patient data file in this study contains patients' personal information such as first and last names, patient ID, and birth date. It also entails information related to the games played such as the session ID, name of the game, movement ID, movement name, and associated dates (*Wilson et al., 2017*). Each selected game and movement acts as one exergame with its unique input variables in the item settings dialogue. The settings include the sides used (left or right), duration, difficulty, tolerance, minimum and maximum ranges. The values of these variables could be fixed based on the default values or adjusted by the physiotherapist after evaluating the performance of the patient. The MIRA platform could generate 26 variables based on the exergame or cognigame (a game that trains the cognitive function). Table 1 describes the most significant variables generated by the exergames.

The experimental data contained 3,553 records generated by 61 patients with different types of diagnoses: 41 patients had a stroke, 14 patients had TBI, seven patients had SCI, one patient had CP, and two patients had humerus. Patients provided written informed consent before the start of each experiment. Fig. 2 portrays an example of the MIRA setting for animals exergame with the elbow flexion movement. The item settings includes six variables that can be manipulated by the physiotherapist or player. Table 2 presents a description of the generated exergame features by patients using the MIRA platform. In each session, a patient plays an exergame by moving his/her limbs according to the rules of the game and movement exercise. During the exercise, the

**Table 1 Description of the output variables by the MIRA platform.**

| No | MIRA application data | Description |
|---|---|---|
| 1 | Time (Duration) | Total time of the exergame item (a game and a movement). |
| 2 | Still time | The idle time of the item when the patient stops moving before the game finishes. |
| 3 | Moving time | Time of the movement when the patient continues moving correctly and incorrectly throughout a game-time. |
| 4 | Moving time in exercise | The time of the movement when the patient continues moving correctly (as required by the exercise) throughout the exergame-time. |
| 5 | Average acceleration | The average of the positive change rate of the velocity divided by the overall duration of the exergame. |
| 6 | Average deceleration | The average of the negative change rate of the velocity divided by the overall duration of the exergame. |
| 7 | Average accuracy | The accuracy of the movement for each exergame. |
| 8 | Average congruent correct answer reaction time | The movement with the objects in the game during positive reaction time where the reaction time is the positive response time for each event in the game. |
| 9 | Average congruent incorrect answer reaction time | Congruent movement with the objects during a negative reaction time by responding to each event. |
| 10 | Average percentage | Average range of motion that the patient performs during the exercise. |
| 11 | Average speed | The result by dividing the distance with the overall Time duration when the movement is performed correctly. |
| 12 | Average variation | The average interval of range of motion, a patient, performs during an exercise. |
| 13 | Distance | Total distances performed by a specific joint. |
| 14 | Maximum percentage | Maximum range of motion carried out by a patient throughout the exergame. |
| 15 | Minimum percentage | Minimum range of motion carried out by a patient throughout the exergame. |
| 16 | Repetition | The total number of correct movements throughout the exergame. |
| 17 | Points | The total scores achieved throughout the exergame. |

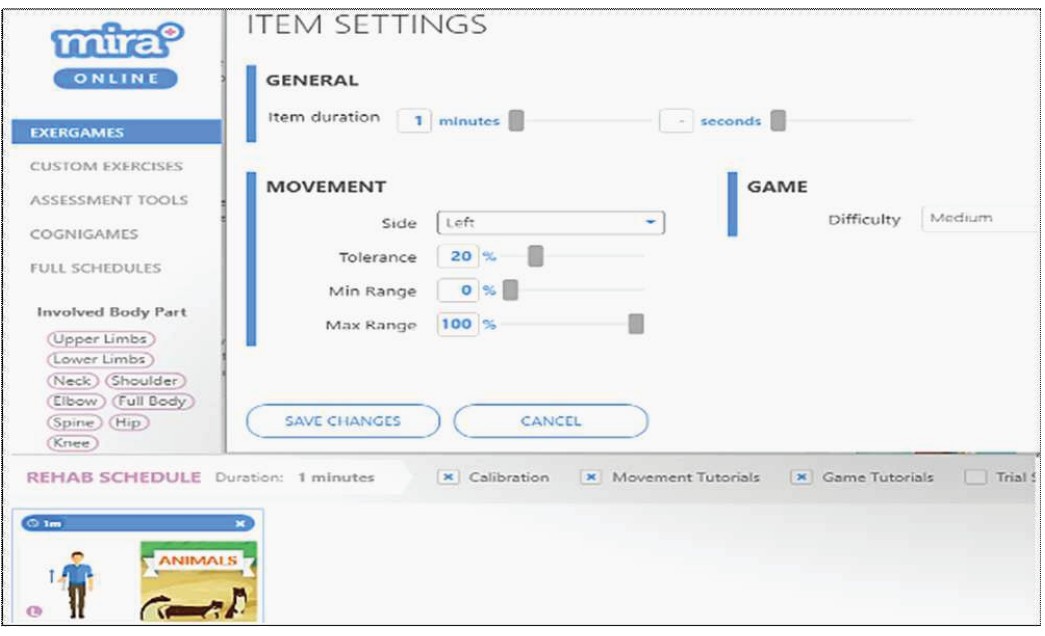

**Figure 2 An example of the dialogue box for item settings.**

**Table 2 Description of MIRA platform settings.**

| No | MIRA platform setting | Description |
|----|----------------------|-------------|
| 1 | Item duration | Assign the duration of the exergame according to the patient's ability between 1 and 10 min (1 min is the default duration). |
| 2 | Side | Assign the side of the patient used in the exergame, which can be left or right, or even both. |
| 3 | Tolerance | Assign the percentage between 0% and 100%. The tolerance is the accepted error in performing the required exercise, to enable the patient to play easily when they cannot perform the entire correct movement of the exercise. |
| 4 | Min range | Assign the percentage between 0% and 100%. The minimum range of motion required by the patient to play exergames. |
| 5 | Max range | Assign the percentage between 0% and 100%. The maximum range of motion required by the patient who plays an exergame. |
| 6 | Difficulty of game | Assign the difficulty of the exergame with values easy, medium or hard. |

physiotherapist predicts the variables of the input setting such as difficulty, tolerance, minimum range, and maximum range according to his observation or adopts the default values, but a number of patients experienced difficulty playing the games. Afterwards, he deduced the accurate settings from the previous performance of patients in each exergame. As compared to using the default settings, a more accurate setting ensures patients play better. This indicates the significance of this research to the MIRA platform.

In this study, the ReComS approach is proposed to predict the variables of the input setting according to the data history of the patient. As ReComS is expected to provide low prediction accuracy, it is integrated with a clustering method (as used in similar experimental works (*Al-Hadi et al., 2017*, *2020a*)) and referred to as the ReComS+ approach. ReComS+ provides good prediction accuracy. ReComS++ is developed by further integrating ReComS+ with the BFOA algorithm to learn the latent features of the patients and to lower the RMSE value throughout the learning iteration process. The experimental results of ReComS and ReComS+ are utilised to benchmark the prediction performance of the ReComS++ approach.

## ReComS approach

Most personal recommendation systems use the CF and the K-NN for providing personal recommendations. Here, the CF technique provides the target patient with personal recommendations according to the common behaviours of other patients. K-NN method is used to obtain the nearest neighbours of each target patient based on their similarities (*Portugal, Alencar & Cowan, 2018*). Thus, ReComS integrates the K-NN algorithm with the CF technique for learning the personal behaviour of patients and predicting the input setting variables. The proposed ReComS approach assists the physiotherapist to collect accurate data from patients who need to play exergames using MIRA platform.

The framework of ReComS is arranged following the steps in Fig. 3. The ReComS approach is set to the target patient to manage the entire patients' features in the features matrix. K-NN is applied to classify the $k$ nearest neighbours based on the similarities between the target patient and other patients. The RMSE function calculates ReComS prediction accuracy based on the distance between the features of the target patient and prediction values obtained using this approach.

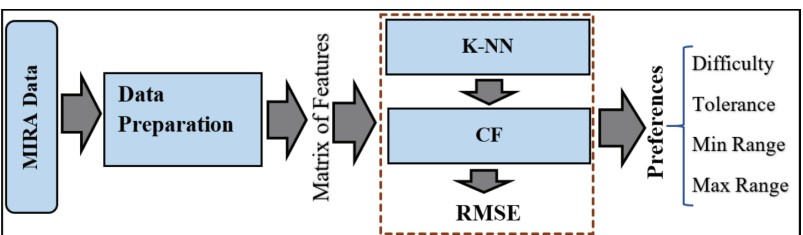

**Figure 3  The framework of the ReComs approach.**   

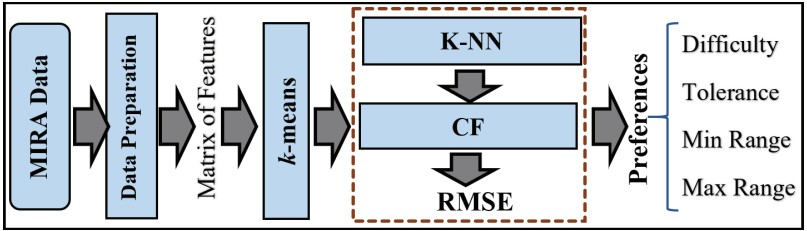

**Figure 4  The framework of the ReComs+ approach.**   

## ReComS+ approach

ReComS+ is proposed to improve the prediction accuracy of ReComS. The ReComS approach should yield higher accuracy, even if the RMSE remains high. $k$-means, K-NN, and CF methods are integrated into ReComS+ to provide the predicted variables as input setting in the MIRA dialogue box of each exergame. The $k$-means algorithm clustered the records of patients into the sum of clusters ($k$). The cluster that contained the record of the target patient is selected to obtain the matrix of neighbours that are integrated with the CF to provide the preferences values, as shown in Fig. 4.

## ReComS++ approach

Despite the high accuracy of ReComS+ prediction, its error value is relatively high. This error value can be reduced by deploying learning methods. Thus, BFOA is integrated with the ReComS+ for learning the behaviours of neighbours by lowering the error value during the iteration stages. The framework of ReComS++ encapsulates four stages that are needed to provide the preferences for patients' input settings, as shown in Fig. 5. These stages are described as follows:

### Data preparation

- Reading data records and arranging variables in a matrix.
- Encoding the textual data (such as gender, diagnosis, game name, difficulty and side) using numbers.
- Normalising the generated variables of each exergame based on the duration by applying Eq. (5).

$$F = F * T_d / T_F, \tag{5}$$

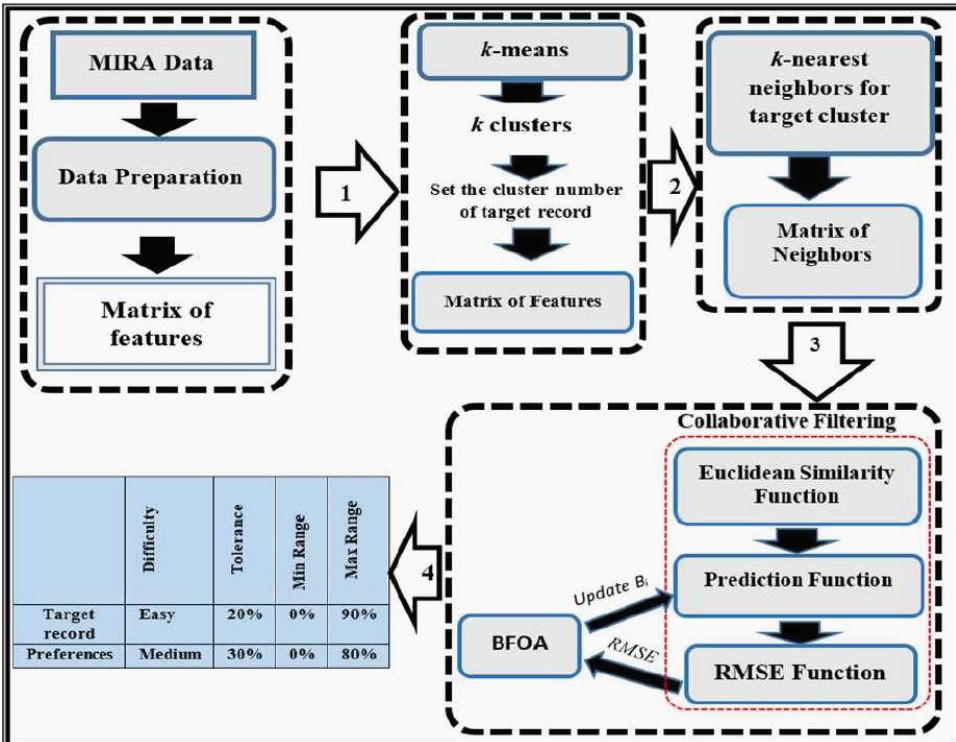

**Figure 5 The framework of the ReComs++ approach.**

where $F$ is a feature value, $T_d$ is a default duration i.e., 60 s, in this process, and $T_F$ is the duration of the exergame. Eq. (5) is implemented for the variables whose values increase with duration. These include Time, Moving Time, Moving Time in the Exercise, Still or Idle Time, Distance, Points, and Repetition. Other variables do not increase with duration since they are considered as either average values or percentage values between 0 and 100.

- Normalising the variables into the range between 0 and 1. This is based on the dimension of each variable, features rescaling and the need to provide proper compatible values for machine learning algorithms. The normalization is performed based on the standard data mining requirement to provide accurate variables approximation and prediction using Eq. (6).

$$F_{ij} = \frac{F_{ij} - X}{Y - X}(\partial - \varnothing) + \varnothing, \tag{6}$$

where $F_{ij}$ is the value of record $i$ and variable $j$, $X$ is the least value, and $Y$ is the highest value in the whole matrix of variables. $\partial$ is the highest target value (1) and $\varnothing$ is the least target value (0).

- Assigning the target patient, target movement, and side, to find the latest record of the patient where he played the selected movement exergame on the target side. Variables of this record would be arranged in the first row in the matrix of features.

- Selecting whole records from data containing the target movement of the target patient and putting the variables of these records in the matrix of features.
- The matrix of features would be divided into two parts. The first part would have 70% of records for the purpose of training and the second part would have 30% for evaluations.

### Clustering by k-means algorithm

The MIRA data consists of 30 games and 34 movements that provide 1,020 kinds of features, which make it challenging to analyse. The features are grouped into a single matrix. Hence, the $k$-means clustering algorithm is used to simplify the various types of features generated by playing the games and movements in the MIRA application. These features are divided into a set of clusters. The challenge of this experimental work is in determining the accurate sum of clusters. To address this issue, MIRA data are tested in a set of $k$ clusters ranging between 5 and 10. Based on the data collected, we assign the range from 5 to 10 clusters as the sufficient range of clusters. This is intended to avoid the $k$-means clustering problem ($k$ problem) when using clusters above 10 because of the high number of zeros in each matrix of features. After that, the prediction performed by ReComS is assessed based on this set of clusters.

### Classification by K-NN algorithm

The K-NN is applied to retrieve similar records to the target record in the matrix. The challenge in this process is in determining the accurate sum of neighbours. A small number of neighbours yields an accurate prediction performance while a vast number of neighbours exhibits the lowest performance. However, the few numbers of neighbours are not sufficient to learn the accurate features of patients. Thus, a larger number is required to accurately learn the patients' features. Addressing this problem, BFOA is integrated with the ReComS+ to improve the prediction accuracy. In this experimental work, the K-NN algorithm applies the squared ED, as given in Eq. (7).

$$D(x - y) = \sqrt{\sum_{i=1}^{n} (xi - yi)^2},$$

(7)

where $D$ refers to the distance value between records $x$ and $y$ while $n$ denotes the number of features. Eq. (7) calculates the distance between the target record and the total records of the target cluster. The target record is hinged on the target cluster mainly because the distance between the target record and the centroid point of this cluster is smaller when compared to other clusters. After calculating the distance between the total records of the target cluster and the target record, the values are sorted in ascending order to derive closely related neighbours to the target record. Records having small values of distance have the highest similarity value to the target record thus, being the nearest neighbours. In this experimental work, the $k$ of neighbours is determined based on a set of $k$, i.e., 25 neighbours, 50 neighbours, 75 neighbours and 100 neighbours. These four numbers of neighbours have been chosen according to the available features of each exergame within each cluster. These features constitute an effective solution for executing the required

training processes for three reasons. First, most patients like to play only a few interesting MIRA exergames. Thus, other exergames have fewer records. Meanwhile, the machine learning algorithms need a large number of records to facilitate the process of learning the latent-features using the exergame output features. Second, interesting exergames can be predicted easily due to their rich output clusters using most similar features to the target exergame. Then, K-NN method can find $k$ neighbours of over 100 records while it is more difficult in clusters with less than or equal 100 records. Third, some patients need to play specific exergames to improve their idle movement skills. The number of output records of these specific exergames is small. Thus, the output cluster of the target matrix of such special exergame does not converge due to the gap between them and the popular exergames. Hence, there is only a few neighbours such as 25 or 50 from these output clusters while it is quite difficult to find 75 or 100 neighbours. In addition, more than 100 neighbours can be considered as impossible or inaccurate due to the resulting poor convergence between the features of target exergame and those of poor-performing clusters. Each $k$ neighbour is then evaluated using ReComS based on its prediction performance in comparison to other numbers of neighbours.

### Developing CF performance with BFOA

The notion of predicting values for variables that constitute the item settings in MIRA, based on the behaviour of patients that have played a few exergames, is similar to the idea of predicting products for customers based on their preferences in the recommendation system that uses the CF technique. Obviously, the CF predicts the score values for products while ReComS predicts variables for all input and output features associated with a specific game and a peculiar movement. Typically, the CF technique uses three functions to estimate values, as described next.

- Similarity function

   This function provides the correlation between the target record (of game and movement) and total records. The similarity functions that apply the CF technique are Cosine Similarity (*Al-Hadi et al., 2020b*) and the Pearson Correlation Coefficient (*Srifi et al., 2020*). Note that when using the Cosine or Correlation coefficient for MIRA data, these functions generate some outliers due to the existence of zeros in the feature matrix. Thus, the Euclidean similarity function, as shown in Eq. (7), emerges as the most suitable similarity function to be applied for MIRA data, mainly because all the calculated similarity values are known.

- Prediction function

   This is an important computational procedure obtained from the similarity values retrieved from the similarity function and the correlation between the total records. For the purpose of prediction in this experimental work, Eq. (8) has been proposed based on the current prediction function in the CF technique (*Natarajan et al., 2020*) after considering the difference between rating scores and features values of MIRA.

**Peer**J Computer Science

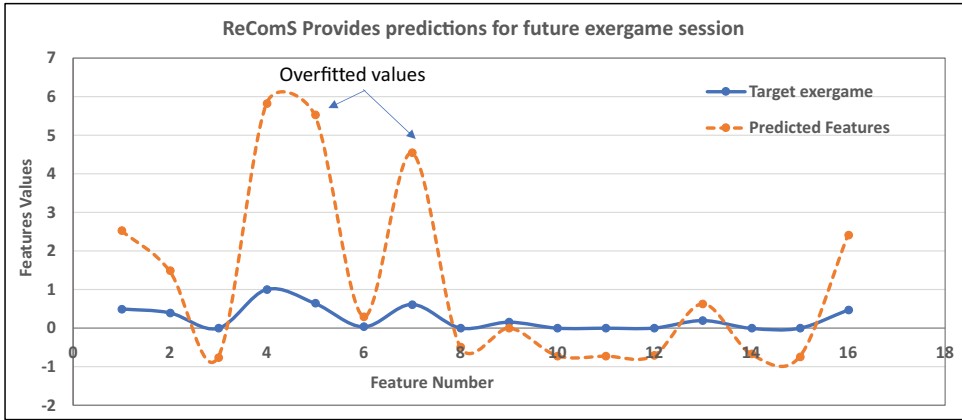

**Figure 6 An example of the overfitted predicted features using ReComS.**

$$P_i = V_a + \frac{\sum_{h=1}^{N} D(F_a, F_h)(F_{h,i} - V_h)}{\sqrt{\sum_{h=1}^{N} D(F_a, F_h)}}, \tag{8}$$

where $P_i$ is the predicted or projected value for feature $i$, $V_a$ is the average value of all feature values for the target record, $N$ is the sum of neighbours, $D$ is the distance similarity value between $F_a$ (feature value of target record) and $F_h$ (feature value of neighbour $h$). Also, $F_{h,i}$ refers to the feature value $i$ of the neighbour $h$, whereas $V_h$ denotes the average value of all features of neighbour $h$.

In this work, Eq. (8) is used in ReComS and ReComS+ by employing the error function. Nevertheless, the generated output still has errors and the predicted values are over-fitted. Such over-fitting occurs when the predicted value is larger than the features generated by the target exergames. Fig. 6 graphically exemplifies the generated features of the target exergame and the features predicted by the CF technique within the procedures of the ReComS approach.

In Fig. 6, eight predicted feature values are overfitted because these values are greater than the feature values of the target exergames. The remaining predicted features have lower fitting values due to their values are smaller than the feature values of the target exergames. Hence, the predicted features need to be normalised to fit/align these values with the target exergame features. Nonetheless, the ReComS and ReComS+ are inaccurate approaches in normalizing the predicted features. Thus, an optimised algorithm is embedded in the prediction method to normalise the prediction values. Notably, BFOA has been acknowledged as an optimisation algorithm commonly applied in recommendation systems (*Hwangbo, Kim & Cha, 2018*) since this algorithm can exceptionally learn the deep features of each matrix. The contribution of the ReComS++ approach is represented in Eq. (9).

$$P_i = V_a + \frac{B_i \sum_{h=1}^{N} D(F_a, F_h)(F_{h,i} - V_h)}{\sqrt{\sum_{h=1}^{N} D(F_a, F_h)}}, \tag{9}$$

**Table 3 The factors' values of BFOA.**

| BFOA factors | No | Parameters | No |
|---|---|---|---|
| P dimension of search space | 30 | Reproduction steps | 4 |
| Number of bacteria groups $S$ | 6 | Elimination-dispersal steps | 4 |
| Number of iterations | 20 | Probability of elimination-dispersal | 0.25 |
| Optimum RMSE | 0.1 | $d_{attract}$ | 0.1 |
| Run length unit $C_i$ | 0.09 | $w_{attract}$ | 0.2 |
| chemotactic steps | 6 | $h_{repellant}$ | 0.1 |
| The swimming length | 4 | $w_{repellant}$ | 5 |

where $B_i$ is the bacteria value that can be learned by tracking feature $F_i$, and the sum of bacterium members will be equal to the number of neighbours' features. These bacteria have been used to track all features of the neighbours (such that each column in the matrix is managed by a bacterium member) and provide accurately predicted input variables for MIRA. The remaining vectors of Eq. (5) have been described in Eq. (4). The BFOA is implemented based on the algorithmic phases described in "Bacterial Foraging Optimisation Algorithm" while the values of bacteria factors are listed in Table 3.

- Benchmark function

The proposed approaches are evaluated to examine the performance of the CF technique using RMSE (*Al-Hadi et al., 2020a*) and Mean Absolute Error (MAE) (*Wang, Yih & Ventresca, 2020*). In this article, RMSE measure is used for calculating the differences between the variable of target patient and predicted values for same variables of target patient. These variables are the input setting variables (difficulty, tolerance, minimum range, and maximum range) and the generated variables by exergame such as average acceleration, average deceleration, moving time and other variables that are described in Table 1. The predicted values are calculated by Eq. (9) according to the similarity values for patients comparing to the target patient variables.

In other words, the total number of generated exergame features of all patients are divided into $k$-clusters using the $k$-means. Each cluster is classified by the K-NN algorithm that selects the most suitable features needed to provide accurate feedback. BFOA focuses on accurately learning the latent features. This is achieved by tracking the positive effects produced through classified features by reducing the RMSE value throughout the optimization stages. This is achieved by accurately learning the convergence among the generated features for various patients who played various exergames, as shown in Eq. (10).

$$RMSE = \frac{1}{R}\sum_{p=1}^{R}\sqrt{\frac{1}{n}\sum_{i=1}^{n}\left(F_{a,i} - P_i\right)} \,, \qquad (10)$$

where *RMSE* is the average *RMSE* values for all records in the matrix of neighbours, $R$ denotes the number of records in the training or testing sets, $n$ refers to the number of

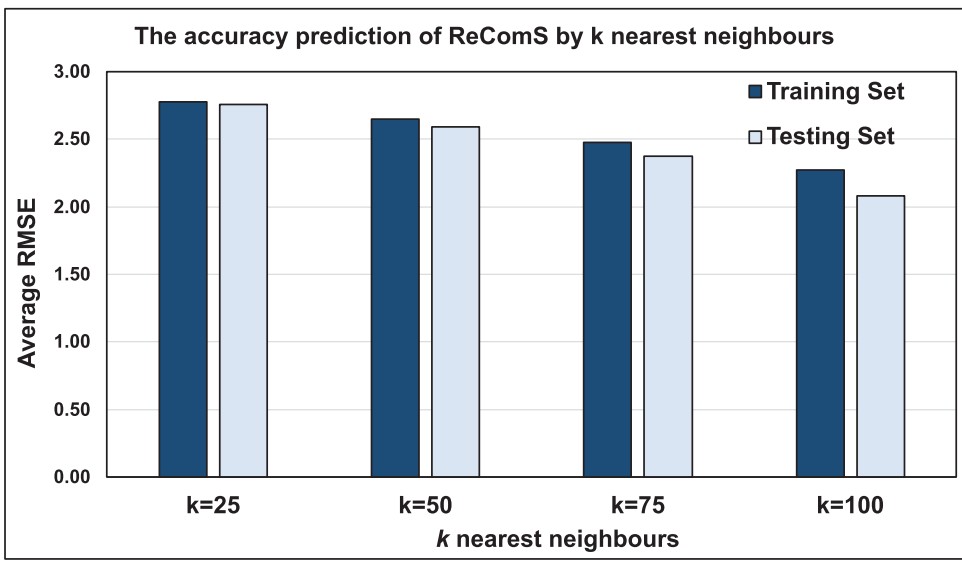

**Figure 7  The prediction performance (RMSE) of ReComS.**

features in the matrix, $F_{a,i}$ represents the value of feature $i$ in the target record, and $P_i$ stands for the predicted value recommended to feature $F_i$. The highest RMSE value reflects the lowest accuracy prediction performance.

## RESULTS

ReComS improves the performances of the following three experimental procedures.

### Evaluating the ReComS approach

The K-NN algorithm classifies integral features of a vast number of patients who played several games and with varying movements. CF computes the similarity between the target features and the features of each neighbour before predicting the new values. The performance of the predicted values is consequently assessed using the RMSE value. To benchmark the varied RMSE values for several target records based on patients' personal behaviours, the average RMSE is calculated to obtain accurate outputs. Fig. 7 illustrates the prediction performance accuracy using this approach based on two sets, training and testing sets. Both sets show that 25 nearest neighbours yields the lowest performance accuracy when compared to those having 50, 75, and 100 neighbours. The case of 100 neighbours yielded more accurate performance when compared to that of 25 neighbours. This deteriorates the prediction performance of the nearest neighbours, which may be solved by other classification approaches.

### Evaluating the ReComS+ approach

The $k$-means clustering algorithm is applied in ReComS+ to address the limitation of ReComS using CF and K-NN. Deploying these methods, even the use of a small number of nearest neighbours could result in a highly accurate performance. Fig. 8 indicates that after integrating the $k$-means algorithm into ReComS, the prediction performance of CF is

**Peer**J Computer Science

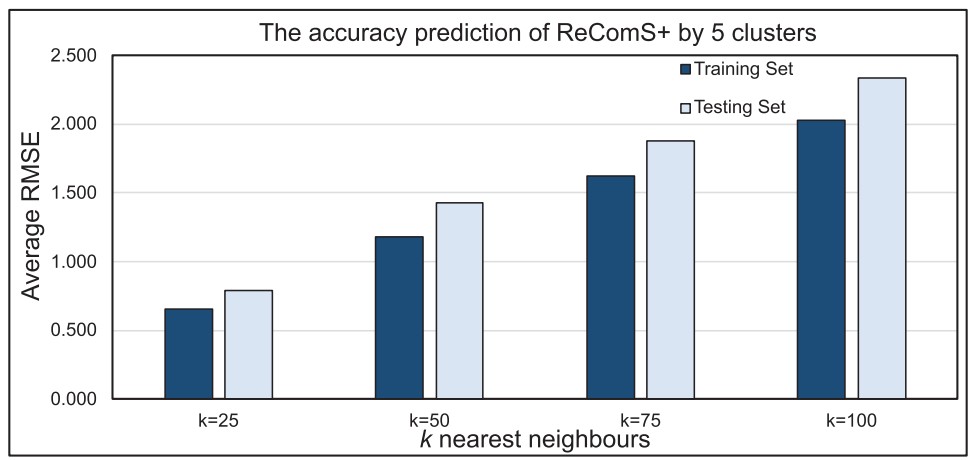

**Figure 8  The prediction performance of ReComS+ using five clusters.**

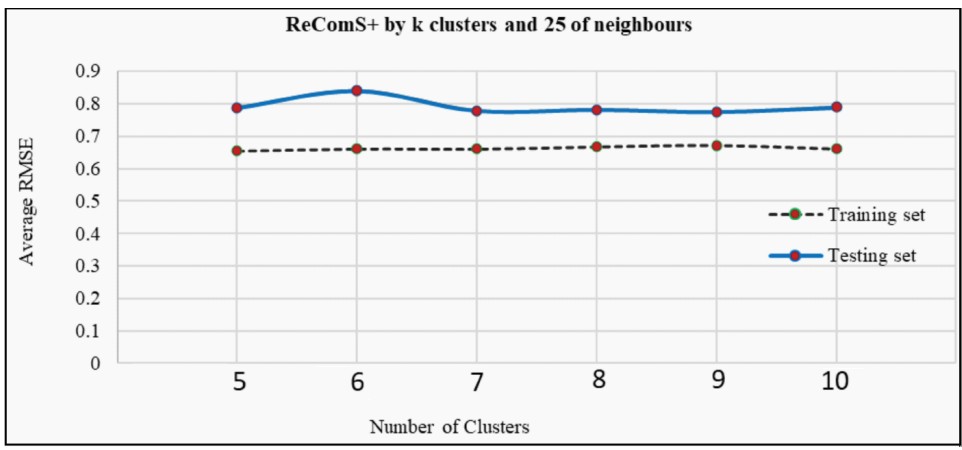

**Figure 9  The prediction performance (RMSE) of ReComS+ using various clusters.**

improved and the feedback of the nearest neighbours are corrected. The outcomes, as depicted in Fig. 8, show the prediction performance of CF using five clusters and 25 neighbours is better for both the training and testing sets, as compared to the results when 50, 75, and 100 neighbours are integrated with five clusters. Nevertheless, the number of clusters is not justified, as six or ten clusters may offer a higher prediction accuracy as shown in Fig. 9. Hence, ReComS+ tests the prediction performance using $k$ clusters to address the problem associated with cluster numbers.

In addition, Fig. 9 proves that various numbers of clusters can provide similar prediction performance as ReComS+ for both the training and testing sets. The results show that five clusters provide the highest accuracy prediction using the training dataset when compared to the performance of ReComS+ by the other $k$ clusters, which ranged between six and 10. The accurate prediction of ReComS+ in the testing dataset, therefore, confirms that the prediction performance of ReComS+ for all $k$ clusters is similar to that of

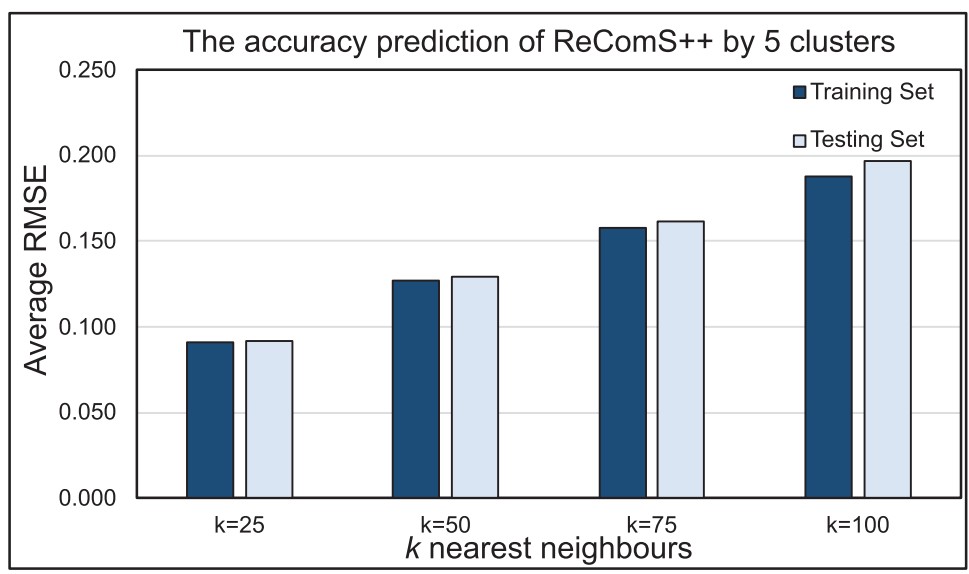

**Figure 10 The prediction performance of the ReComS++ approach based on optimisation.**

6 clusters that generate low accuracy performance. For this reason, subsequent experimental works using ReComS++ for five clusters is required. Though the accuracy performance of ReComS+ has been improved using this approach, the RMSE is still high and the range of the predicted values should be normalised. For this reason, the BFOA is applied to normalise the predicted values and minimise RMSE values.

### Evaluating ReComS++ approach

BFOA is implemented in this work to normalise the predicted values of the ReComS+ approach, which utilised the CF, K-NN and $k$-means methods. The results proved that the predicted values are similar to the variables of the target record while emphasizing the need to decrease RMSE values. Figure 10 presents the prediction performance of the ReComS++ approach that employed CF, K-NN, $k$-means, and BFOA for both training and testing sets. The RMSE values, through this approach, appeared to be small, thus indicating high prediction accuracy for all sets of tested neighbours. The set of 25 neighbours provide the highest prediction accuracy when compared to the other sets of neighbours (i.e., 50, 70 and 100) for the MIRA training and testing datasets. The results of both training and testing sets are close, indicating that this approach provides accurate prediction values for the whole target records in the MIRA data.

### DISCUSSION AND FUTURE WORKS

The CF technique is applied to predict the values of future variables of the item settings. This technique is integrated with the K-NN algorithm in ReComS. It provides each exergame with predicted feature values related to the generated features of the target exergame. Few of these predicted features can be used to assign the variables of the exergame dialogue box setting. The ReComS approach provides a low prediction accuracy due to the high percentage of the overfitted predicted values. Hence, ReComS needs further

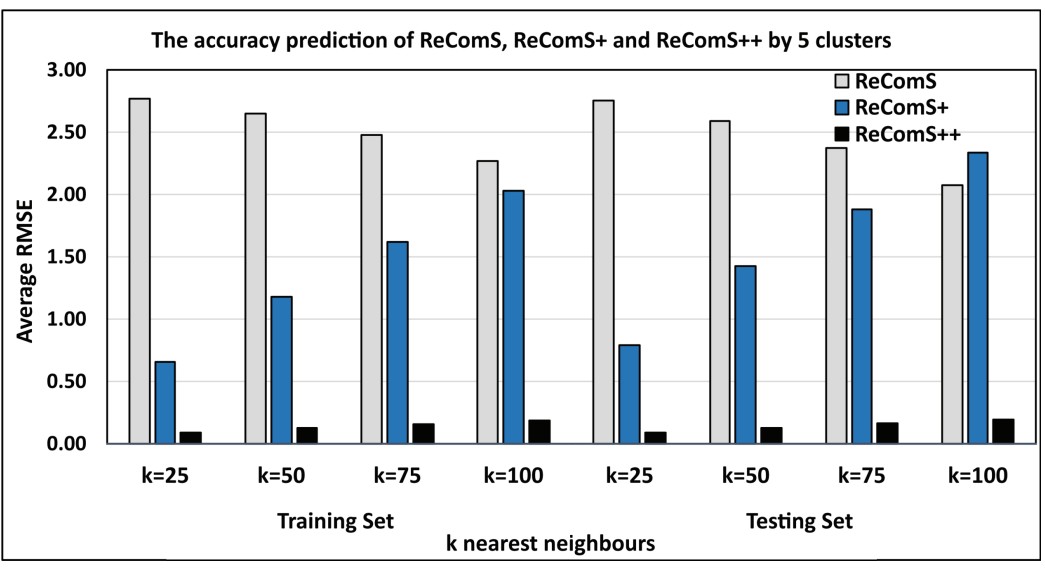

**Figure 11 Comparisons among the ReComS approaches.**

improvement to classify the various output features of all exergames. For this reason, ReComS is subsequently improved by ReComS+ that utilizes the $k$-means algorithm for grouping the various generated features into $k$ clusters and then finding the nearest features to the target exergame features within the same cluster.

ReComS+ approach chooses the best cluster and number of neighbours through its accurate predictions. The prediction performance of ReComS+ is more accurate than that of ReComS because the former accurately learns the convergence among exergame features using the $k$-means. Similarly, each cluster is classified to accurately learn the neighbours' features using K-NN. Nevertheless, the predicted values generated in ReComS + vary and the prediction accuracy of ReComS+ is low due to the difficulty in learning the latent features of the vast amount of generated exergame features. Thus, the ReComS+ approach should be further improved using an optimization algorithm that can effectively learn the latent features of the neighbours within each cluster.

BFOA is one of the efficient optimisation algorithms that have been used in improving the prediction performance of the CF technique in some recommendation systems (*Al-Hadi et al., 2020b*, *2017*). Accordingly, BFOA is utilized in the ReComS++ approach for reducing the overfitted prediction values by learning the latent features of the neighbours within each cluster. Further, BFOA is used to ensure the outlier data belonging to the cluster. The experimental approaches show that the ReComS++ approach has addressed the inherent challenges of the first and second approaches. Figure 11 shows the comparisons between the RMSE values of the three experimental approaches: ReComS, ReComS+, and ReComS++ for MIRA training set. ReComS++ approach provides the lowest RMSE value, indicating it has highest prediction accuracy when compared with both ReComS and ReComS+ approaches. Furthermore, Fig. 11 illustrates the experimental outcomes derived from the MIRA testing dataset. The results are similar to those

|  | Difficulty | Tolerance | Min Range | Max Range |
|---|---|---|---|---|
| Min Predicted Value | -0.336 | -0.173 | -0.378 | 0.167 |
| Max Predicted Value | 1.38 | 1.360 | 1.250 | 1.461 |

| Difficulty values (D) | $D < 0.40$ | $D >= 0.40$ & $D <= 0.70$ | $D > 0.70$ |  |  |
|---|---|---|---|---|---|
| Predicted Difficulty | Easy | Medium | Hard |  |  |
| Tolerance values (T) | $T < 0.3$ | $T > 0.30$ && $T < 0.40$ | $T > 0.40$ & $T <= 0.60$ | $T > 0.60$ & $T <= 1.0$ | $T > 1$ |
| Predicted Tolerance | 20% | 30% | 40% | 50% | 60% |
| Min Range values (M) | $M <= 0.8$ | $M > 0.80$ && $M <= 1.20$ | $M > 1.20$ |  |  |
| Predicted Min Range | 0% | 10% | 20% |  |  |
| Max Range values (Mx) | $Mx <= 0.6$ | $Mx > 0.60$ && $Mx < 0.80$ | $Mx >= 0.8$ && $Mx <= 1.0$ | $Mx > 1.0$ & $Mx <= 1.20$ | $Mx > 1.20$ |
| Predicted Max Range | 60% | 70% | 80% | 90% | 100% |

**Figure 12 The threshold for deciding the predicted variables' values of the item setting in MIRA.**

generated for the training dataset using the ReComS++ approach. This indicates that ReComS++ successfully addressed the drawbacks of the ReComS and ReComS+ approaches.

There are several potential promising directions where ReComS++ approach can be integrated with other platforms that have the profile settings for exergames. It may be used to learn latent features of the exergames output features for any platform that has settings variable for predicting the input settings' variables. Further, ReComS++ approach provides that 85% of predicted results are correct, while this result can be upgraded up to 90% by exploring other machine learning methods to reduce the computational time of ReComS++ approach throughout the iterative learning. It can focus on specific variables such as average correct answer reaction time for cognitive games. The cognitive and the repetition of range of motion need more study to understand the suddenly patient movements and predicting the suitable settings. Therefore, we intend to explore various linear regression techniques for predicting the optimal setting variables. In addition, we plan to explore several structures of deep learning methods to find the best model that can reduce the computational time and learn more accurate latent features of patients based on their personal behaviours from playing the exergames.

## Criteria for the ReComS++ approach according to the output

A significant milestone in this work is determining the predicted difficulty level and the remaining variables in the dialogue box for the setting of each item. Figure 12 shows the minimum and maximum predicted values for the difficulty, tolerance, as well as minimum and maximum ranges. Based on the predicted and observed values obtained

| MIRA Application | Variables of item setting | Difficulty | Tolerance | Min Range | Max Range |
|---|---|---|---|---|---|
| Target Record of item | Actual values in data of MIRA | Easy | 30% | 0% | 60% |
| | Values by Normalisation | 0 | 0.3 | 0 | 0.6 |
| Predicted Values of the future item | Actual predicted values | -0.075 | 0.221 | -0.086 | 0.759 |
| | Predicted Values by Threshold | Easy | 20% | 0% | 70% |
| Target Record of item | Actual values in data of MIRA | Medium | 100% | 0% | 100% |
| | Values by Normalisation | 0.5 | 1 | 0 | 1 |
| Predicted Values of the future item | Actual predicted values | 0.823 | 1.035 | 0.188 | 0.897 |
| | Predicted Values by Threshold | Hard | 60% | 0% | 80% |

**Figure 13 An example of the output predicted values for the item setting by the ReComS approach.**

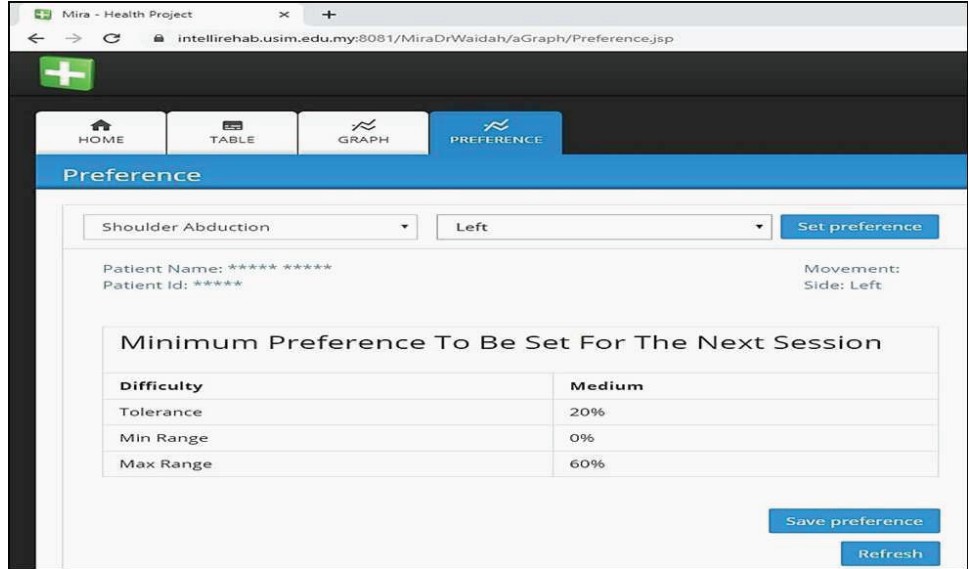

**Figure 14 MIRA interface provides preferences by ReComS++ for the input settings.**

from the MIRA application in Melaka, Malaysia while supervising patients who played MIRA games. The determined threshold intervals are as illustrated in Fig. 13 where this figure presents two observations for two target records created by the output of the predicted variables using ReComS++. First, the example in Fig. 13 depicts the actual observation made by the physiotherapist for a patient who has played the game with movement exercise at an easy (difficulty) level with a tolerance up to 30% (percentage of range of movement is 0–60%). After that, the experimental approach normalises these values into the range between 0 and 1, as shown in the figure. The experimental approach performs closely to the actual values in terms of difficulty, tolerance, minimum range, and maximum range. Based on the threshold shown in Fig. 12, the experimental approach

**Table 4 An example of information collected by the physiotherapist for MIRA and ReComS++.**

| Date | Patient ID | Movement | Side | Preference | | | | Game | Observation |
|------|-----------|----------|------|-----------|---|---|---|------|-------------|
| | | | | Tolerance | Min Range | Max range | Difficulty | | |
| 5/8/2019 | 3522 | Elbow Flexion | R | 20 | 0 | 100 | Easy | Catch | P |
| | | General Arm | R | 20 | 0 | 80 | Easy | Catch | P |
| | | Shoulder Internal Rotation | R | 20 | 0 | 70 | Easy | Catch | P |
| | | Shoulder External Rotation | R | 20 | 0 | 100 | Easy | Catch | P |
| 5/8/2019 | 3566 | Elbow Flexion with Abduction | L | 30 | 0 | 100 | Easy | Colour Blocks | P |
| | | General Full Body | L | 20 | 0 | 100 | Easy | Basketball | P |
| | | Shoulder Abduction | L | 20 | 0 | 100 | Easy | Colour Blocks | P |
| 5/8/2019 | 3597 | Elbow Flexion | L | 20 | 0 | 100 | Medium mm | Catch | N |
| | | Spine Lateral Flexion | L | 20 | 0 | 100 | Medium | Colour Blocks | P |
| | | Spine Frontal Flexion | L | 20 | 0 | 100 | Medium | Catch | P |
| | | Shoulder Internal Rotation | L | 20 | 0 | 100 | Medium | Catch | P |
| 5/8/2019 | 3553 | Elbow Flexion | L | 30 | 0 | 100 | Easy | Catch | P |
| | | Spine Frontal Flexion | L | 20 | 0 | 70 | Easy | Catch | P |
| | | Spine Lateral Flexion | L | 30 | 0 | 100 | Easy | Colour Blocks | P |
| | | Functional Reach | L | 30 | 0 | 70 | Easy | Grab | P |
| | | Shoulder Frontal Flexion | L | 30 | 0 | 100 | Easy | Colour Blocks | P |
| | | Sit To Stand | L | 20 | 0 | 100 | Medium | Atlantis | P |
| 5/8/2019 | 3598 | Elbow Flexion in Abduction | L | 20 | 0 | 100 | Medium | Colour Blocks | P |
| | | General Full Body | L | 20 | 0 | 100 | Medium | Basketball | P |
| | | Functional Reach | L | 20 | 0 | 100 | Medium | Grab | P |
| 6/8/2019 | 3501 | General Arm | R | 20 | 0 | 100 | Easy | Catch | P |
| | | General Shoulder | R | 20 | 0 | 100 | Medium | Catch | P |
| | | Elbow Flexion | L | 20 | 0 | 100 | Easy | Catch | N |
| | | Spine Lateral Flexion | | 30 | 0 | 100 | Easy | Colour Blocks | P |
| 6/8/2019 | 3540 | Elbow Flexion | L | 20 | 0 | 100 | Medium | Catch | P |
| | | Shoulder Internal Rotation | L | 20 | 0 | 100 | Medium | Catch | P |
| | | General Arm | L | 20 | 0 | 100 | Medium | Catch | P |
| | | General Arm | R | 20 | 0 | 100 | Medium | Catch | P |

determines the final prediction values of the next item settings as Easy; 20 for tolerance, 0 for min range and 70% for max range. The second example depicts similar procedures needed to arrive at the final decision in accordance with the threshold.

## Evaluating the ReComS++ approach based on the physiotherapist observations

The ReComS++ approach, programmed using Java, is included in the MIRA system to provide physiotherapists and patients with preferences. Figure 14 shows the interface of the preferences where the physiotherapist (who helps patients to play MIRA exergames) could easily obtain the recommended preferences for the selected patient, movement, and side. On evaluating the ReComS++ approach, we obtain the evaluation file completed

**Table 5 Effectiveness of preferences by ReComS++ according to the physiotherapist's observation.**

| Positive preferences | Number | Results by percentage |
|---|---|---|
| Patients have played a set of exergames by MIRA | 28 | |
| Period of evaluation | 5 weeks | |
| Exergames preferences | 1,182 | |
| Negative preferences (N) | 168 | 14.21% |
| Positive preferences (P) | 1,014 | 85.79% |

by physiotherapists of the Perkeso Rehabilitation Centre in Melaka, Malaysia. The file records the physiotherapists' observations after using the preferences suggested by ReComS++ for a set of patients over a period of 5 weeks. The file contains 1182 records. Table 4 reveals an example of the information provided in the file.

Table 4 presents the set of movements performed by a group of patients who play a set of exergames (movements and games) using the MIRA platform. Each exergame has four preferences that constitute the input setting variables in ReComS++ (i.e., difficulty, tolerance, minimum range and maximum range). Here, the physiotherapist observes each patient who performs the exergame and registers his/her activity performance as positive (P) or Negative (N). The evaluation results are summarised in Table 5. The table shows a higher percentage of positive preferences compared to negative preferences. This implies that ReComS effectively recommends accurate preferences for patients.

## CONCLUSION

In most cases, patients train their disabled limbs by utilising the facilities offered at rehabilitation centres to regain their limbs functionality. The VRT, such as MIRA, refers to a contemporary rehabilitation technique that aids patients to perform "game-aided" exercises in order to increase their motivation and engagement in physical therapy. Nonetheless, physiotherapists who deal with this application need to predict the values of the input variables of the item settings for each patient manually, which is the main challenge in this domain. Therefore, in this study, we utilise a recommender system to suggest the most suitable settings for patients' movements based on their movement history. Since the exergames generate various features, automated analysis is required to provide a summary of the patient's (movement) performance. To address these challenges, three experimental approaches: (1) ReComS with the CF and K-NN approach; (2) ReComS+ with the CF, K-NN; and (3) k-means approach, in addition to ReComS++ with the CF, K-NN, k-means and the BFOA approach; were proposed and their shortcomings were tested by learning procedures. The experimental results demonstrated that ReComS+ yields more accurate predictions when compared with ReComS while ReComS++ achieves a higher accuracy as compared to ReComS+. Overall, ReComS++ performs best for MIRA exergames as it provides MIRA with the most accurate predictions for the input setting dialogue box. It thus assists patients to perform MIRA exergames correctly.

## ACKNOWLEDGEMENTS

Thank you to Directior Dr. Hafez bin Hussain, who helped in carrying out the research at Pusat Rehabilitation Perkeso Sdn Bhd.

### Funding

This work is supported by the Newton-Ungku Omar Fund from Malaysia Industry-Government Group from High Technology (MIGHT) and Grant code USIM/INT-NEWTON/FST/IHRAM/053000/41616. The funders had no role in study design, data collection and analysis, decision to publish, or preparation of the manuscript.

### Grant Disclosures

The following grant information was disclosed by the authors:
Malaysia Industry-Government Group from High Technology (MIGHT): USIM/INT-NEWTON/FST/IHRAM/053000/41616.

### Competing Interests

The authors declare that they have no competing interests.

### Author Contributions

- Waidah Ismail conceived and designed the experiments, performed the experiments, analyzed the data, authored or reviewed drafts of the paper, and approved the final draft.
- Ismail Ahmed Al-Qasem Al-Hadi conceived and designed the experiments, performed the experiments, analyzed the data, performed the computation work, prepared figures and/or tables, and approved the final draft.
- Crina Grosan performed the computation work, prepared figures and/or tables, authored or reviewed drafts of the paper, and approved the final draft.
- Rimuljo Hendradi conceived and designed the experiments, prepared figures and/or tables, authored or reviewed drafts of the paper, and approved the final draft.

### Ethics

The following information was supplied relating to ethical approvals (i.e., approving body and any reference numbers):

The Social Security Organization granted approval to carry out the study within its facilities (Ethical Application Ref: PRPTAR006.-5(42)).

### Data Availability

The raw data, program and the codebook are available in the Supplemental Files.

### Supplemental Information

Supplemental information for this article can be found online at http://dx.doi.org/10.7717/peerj-cs.599#supplemental-information.

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
