# Peer review of "Improving patient rehabilitation performance in exercise games using collaborative filtering approach"

_PeerJ Computer Science, doi:10.7717/peerj-cs.599_

## Round 0.1 · original submission · Major Revisions

Please carefully address the issues raised by the two reviewers.

Reviewer 1 ·

Basic reporting

The paper provides clear definition of concepts, terms and methods (Collaborative Filtering, Kmeans algorithm, etc.).
Sufficient context is provided to understand: 1) why exergames are useful for patients; 2) why rec sys were needed for the MIRA platform, why exergames are useful for patients.
The overall aim of the paper is clear: analysis of patients' movements and use of rec sys to make suggestions on the most appropriate settings for patients' movements.
The structure of the paper is clear; raw data and figures are shared.

However, the paper needs a professional English review (e.g. line 43 "who having" > "who are having"; the overuse of which statements in line 487 and 488, and again in line 69 and 70; in line 47 "where is a" > "where there is a", etc.).

The innovative contributions for MIRA platform are clear (no rec sys before), however the paper needs further work to demonstrate and articulate why MIRA with a rec sys system is innovative compared to the available research ("to the best of the authors' knowledge, addressing this problem has been neglected in previous work" is based on trust, not evidence - we need evidence. Please add references of other works and how your work is innovative compared to others).

Experimental design

As mentioned before: The overall aim of the paper is clear: analysis of patients' movements and use of rec sys to make suggestions on the most appropriate settings for patients' movements.

Research methods are clear both in the abstract and in the Materials and Methods section. Benchmarks are specified, as well as Dataset and equations used.

As mentioned above, it's needed some work to state how the research fills an identified knowledge gap, as there are probably other exergames systems developed by others as well.

Validity of the findings

As mentioned above, the impact is clear for the MIRA platform, but not clear compared to the state of the art of exergames. If the novelty is provided by the use of recommender systems in an unexplored field (exergames), a clear statement is needed. More extensive state of the art research works in exergames are needed. It's also recommended to add in the conclusions a clear statement of how the usage of rec sys in the MIRA platform would benefit the field of exergames and provide a value compared to other works (it's already clear what results were obtained, just add the overall impact compared to the existing literature).

The rationale looks clear: 3 approaches were explored and tested for the recommender system. Each approach used a combination of different methods (knn, Kmeans, bfoa). RMSE was used to measure the accuracy.

Suggestion: line 609 "the experimental results demonstrated that the recoms+ approach appeared better compared to the recoms approach" " please further develop this sentence specifying why (it's mentioned in the discussion of the results, but add a clear conclusive statement here). Also, "it appeared better" is a vague statement -better in which way?. Same for line 610 about Recoms++.

In the conclusion, there is a reference of Recoms+ having "better" experimental results than recoms, and recoms++ providing better accuracy performance than recoms+. Does this means recoms++ is the best approach because the accuracy is better, and is therefore the best approach for MIRA? If yes, please assert so. If not, please discuss why, in the evaluation, accuracy is not the only relevant parameter for identifying the best approach.

Some observations are described in the Discussion section, just make clear conclusive statements on the Conclusion section.

Reviewer 2 ·

Basic reporting

Language needs proof-reading. A professional, native English-speaking proof-reader is required. There are too many language issues to single out, but this proof-reading is needed.

Conceptual clarity needs improving: consistently refer to either patients or users. For example, "to suggest the most appropriate setting for patients to enhance the users' performance" ==> is the user the same as patient?

Experimental design

The explanation for the Bacterial Foraging Optimisation Algorithm needs to be clarified. While the other two algorithms are commonly used for recommenders, this algorithm is not.

Specifically,
(a) why was it selected?
(b) how was it applied --- the current explanation is describing micro-biology, but the use case here was exergame. The explanation needs to reflect the use case.

The authors allude to these points in later sections of the paper, but they need to be first explained when introducing the algorithm.

The number assignment of clusters needs to be clarified:
"MIRA data are tested in a set of k clusters ranging between 5 and 10 clusters"
>> why this test range?

The same for kNN:
"In this experimental work, the k of neighbours is determined based on a set of k (25 neighbours, 50 neighbours, 75 neighbours, 100 neighbours)."
>> why this range?

The inability of normalization to help overfitting needs to be explained:
"prediction values are over-fitting; hence they need to be normalised to fit the prediction values. Nonetheless, the normalisation method is inapt for this process."
>> why inapt?
>> is there a reference applying BFOA in a similar task before? If so, please include.

Validity of the findings

The application of RMSE needs clarification - the authors state, correctly, that the metric is applied "for calculating the differences between the predicted values and the values observed." But what were these values in the current study? Combination of different variables? Which variables?

The study found that the method that combined three techniques was the best. However, this is typically the case for any machine learning situation - ensemble methods outperform single-method approaches. So, the authors need to clarify if there is anything unexpected in this finding, and what it means for future work - could different ensemble techniques be tested against each other?

Additional comments

Discussion should be enhanced with an evaluation of practical implementability of the tested methods. While the most complex method yielded the best performance, is it feasible to implement? If so, under which conditions (and under which other methods could be better).

Discussion should be enhanced by discussing the weaknesses of the current work. What are they? Honestly explicate how the research could be improved.

Discussion should be enhanced by adding directions for future research. What are the next steps? What other methods could be tested? How "good" are the results in terms of practical implementation and how far are they from yielding a substantial positive impact to patients' recovery?

---

## Round 0.2 · Major Revisions

I am very disappointed by the rebuttal. Instead of honestly provide point-by-point answer to each question, the authors simply referred to a block of lines in the paper answer. Unfortunately, what is in the paper failed to answer the questions at all. I would like to give the authors one last chance to actually provide point-by-point response to every single issue raised by the reviewers (primarily by the second reviewer). I will reject the paper if the revised manuscript is not satisfactory.

---

## Round 0.3 · accepted · Accept

I recommend to accept the paper.

Reviewer 1 ·

Basic reporting

English is far much better compared to the the first version of the paper (last time there were some english mistakes that made difficult to follow the entire - easier to read now.
The abstract is clear: it does specify the background, the need, the purpose of the research, methods and results.
The introduction section is comprehensive of background research on rehabilitation tools such as exergames.
The article structure is fine.
Definitions of concepts is included .

Experimental design

Research question well defined, relevant & meaningful:
The abstract is clear: it does specify the background, the need, the purpose of the research, methods and results.
The introduction also states clearly what are the research contribution by the authors compared to available literature.
The MIRA platform is described, including methods used, datasets, and a discussion of all the approaches (ReComS, ReComS+, ReComS++).

Methods described with sufficient detail: Recommender Systems benchmarks has also been investigated and applied to get an estimation of the recommendation accuracy.

Validity of the findings

According to what has been states in the introduction, the paper seems to be novel compared to what's available in the literature regarding exergames.

The conclusion looks good, I would just suggest to make it longer mentioning again the strength of the proposed approach and the lack of similar works in literature.

The provided Raw data can be opened. Might be good to add a README file in the Raw data folder to facilitate the understanding of what each script does.

Additional comments

I'm going to accept the paper.
My only notes are that the conclusion could have been stronger, and that a README file in the raw data package could have been included, explaining what every script does and how to run the entire recommender system (I can see csv files, consent forms, scripts, but no README).